# *OsMLP423* Is a Positive Regulator of Tolerance to Drought and Salt Stresses in Rice

**DOI:** 10.3390/plants11131653

**Published:** 2022-06-23

**Authors:** Zhanmei Zhou, Jiangbo Fan, Jia Zhang, Yanmei Yang, Yifan Zhang, Xiaofei Zan, Xiaohong Li, Jiale Wan, Xiaoling Gao, Rongjun Chen, Zhengjian Huang, Zhengjun Xu, Lihua Li

**Affiliations:** 1Crop Ecophysiology and Cultivation Key Laboratory of Sichuan Province, Rice Research Institute of Sichuan Agricultural University, Chengdu 611130, China; zmfeb1996@163.com (Z.Z.); zjbio96@163.com (J.Z.); yym1578073713@163.com (Y.Y.); lacrimosez@163.com (Y.Z.); 201702193@stu.sicau.edu.cn (X.Z.); lixiao1234562022@163.com (X.L.); jialewan99@163.com (J.W.); myriceworld@hotmail.com (X.G.); chenrj8@aliyun.com (R.C.); phosphate@126.com (Z.H.); 2Chongqing Army Characteristic Medical Center, Chongqing 400000, China; a15608052927@163.com

**Keywords:** rice, abscisic acid, water deficit, salinity, reactive oxygen

## Abstract

Rice (*Oryza sativa* L.) is one of the main food crops for human survival, and its yield is often restricted by abiotic stresses. Drought and soil salinity are among the most damaging abiotic stresses affecting today’s agriculture. Given the importance of abscisic acid (ABA) in plant growth and abiotic stress responses, it is very important to identify new genes involved in ABA signal transduction. We screened a drought-inducing gene containing about 158 amino acid residues from the transcriptome library of rice exposed to drought treatment, and we found ABA-related cis-acting elements and multiple drought-stress-related cis-acting elements in its promoter sequence. The results of real-time PCR showed that *O**sMLP423* was strongly induced by drought and salt stresses. The physiological and biochemical phenotype analysis of transgenic plants confirmed that overexpression of *OsMLP423* enhanced the tolerance to drought and salt stresses in rice. The expression of *OsMLP423*-GFP fusion protein indicated that *OsMLP423* was located in both the cell membrane system and nucleus. Compared with the wild type, the overexpressed *OsMLP423* showed enhanced sensitivity to ABA. Physiological analyses showed that the overexpression of *OsMLP423* may regulate the water loss efficiency and ABA-responsive gene expression of rice plants under drought and salt stresses, and it reduces membrane damage and the accumulation of reactive oxygen species. These results indicate that *OsMLP423* is a positive regulator of drought and salinity tolerance in rice, governing the tolerance of rice to abiotic stresses through an ABA-dependent pathway. Therefore, this study provides a new insight into the physiological and molecular mechanisms of *OsMLP423*-mediated ABA signal transduction participating in drought and salt stresses.

## 1. Introduction

Drought and salinity are two major abiotic stresses responsible for the decline in growth and productivity of crops worldwide [1]. *Oryza sativa* L. (rice) is one of the most cultivated crops (especially in Asia); many genes related to rice drought and salt stresses have been identified, but the molecular mechanisms of rice response and adaptation to these stresses are still unclear. [2].

Water limitation is the main reason for reductions in rice production. According to future climate prediction models, temperatures will increase and the severity of salinization will continue to increase [3,4]. Therefore, increasing the yield of rice and maintaining its stability under a limited water supply is a major challenge for improving food security [5]. As with other crops, the development of drought-tolerant rice varieties mainly depends on improving the ability to capture and transport water. It is necessary to use the available water effectively in carbon assimilation and the allocation of carbohydrates to grains. Therefore, it is very important to explore the genetics related to drought and salt tolerance [6].

Plant hormones are synthesized only in certain cell types, but they are involved in regulating the physiological responses of the entire plant [7]. Plant hormones usually induce or prevent transcriptional regulators through the ubiquitin–proteasome system, allowing plant hormones to rapidly mediate gene expression through a wide range of adaptive responses [8,9,10]. ABA is an important plant hormone involved in plant tolerance to abiotic stresses [11,12]. ABA controls many adaptation responses to stresses, including activation of osmotic regulation genes, ion transport, and changes in root hydraulic conductivity [13,14]. In view of the importance of ABA for plant physiology and development, it is very important to identify new genes involved in ABA signal transduction.

Major latex protein (MLP) was first identified from the latex of the opium poppy (*Papaver somniferum*) [15,16]. OsMLP423 belongs to the MLP subfamily, and MLP (major latex protein) is a member of the disease-related protein family (Bet v1). The Bet v1 family proteins have been reported to participate in the response to biotic and abiotic stresses [17,18]. Our results showed that *OsMLP423*, like other Bet v1 family proteins, seems to be involved in the resistance to abiotic stresses and improves the drought and salt tolerance of transgenic rice. New evidence showed that RCAR1 and PYL5, which contain a Bet v1 domain, bind ABA and activate the ABA signal pathway through direct inhibition of PP2C, which activates the plant stress response [19,20]. Under osmotic stress conditions, such as drought and high salinity, hypertonic signals lead to the accumulation of the ABA, which, in turn, triggers many adaptive responses in plants [21,22,23]. This indicates that MLP protein may be closely related to ABA reactions.

There are no reports about *OsMLP423* (Os04g39150) in rice currently, but there have been many reports on its homologous genes *At1G24020* (Arabidopsis), *Sb06g019320* (sorghum), *GRMZM02g102356* (maize), etc. Our sequence analysis showed that *O**sMLP423* was conserved in different species. The expressions of PR10 (Course-related protein), MLP, and CSBP (Cell division specific binding protein) subfamily proteins were induced by pathogens as well as abiotic stresses such as salt and drought [16]. The study presented here also confirmed that the overexpression of *OsMLP423* plants improved the tolerance of rice to drought and salt and enhanced the sensitivity to ABA. These results indicate that *OsMLP423* is a positive regulator of tolerance to drought and salt in rice via an ABA-dependent pathway. This paper provides a theoretical basis for developing drought-resistant transgenic rice with a potential application value.

## 2. Results

### 2.1. OsMLP423 Sequence Analysis

The *OsMLP423* gene was located on chromosome 4 of rice, with an open reading frame of 474 bp and encoding 157 amino acids. Some homologous genes of *OsMLP423* were analyzed, and their amino acid sequences were compared (Figure 1a). Approximately 1400 bp upstream of the ATG was analyzed by PlantCARE and found to contain multiple elements associated with the stress response (Figure 1b). These elements included CAAT-box (common *cis*-acting elements in promoter and enhancer regions), TATA-box response element, Box-4 (*cis*-acting element involved in light response), ABRE (ABA *cis*-acting element), CGTCA-motif (Jasmonic acid response *cis* element), MBS response element, MBS (MYB binding site involved in drought induction), etc.

### 2.2. OsMLP423 Localizes to the Nucleus and the Membrane System

In order to detect the subcellular localization of *OsMLP423* in plant cells, *OsMLP423* was fused with a GFP reporter gene and transiently expressed in tobacco tissues under the control of a strong 35S promoter. The results showed that the fluorescence signal of the *OsMLP423*-GFP fusion protein was mainly localized in the nucleus and was also detected in the membrane system (Figure 2b).

### 2.3. GUS Staining of Promoter Transgenic Plants

To confirm the tissue-specific expression pattern of OsMLP423, we analyzed the activity of β-glucuronidase (GUS) in transgenic plants controlled by the promoter of OsMLP423. We found that OsMLP423 was expressed in stems, leaves, and spikes (Figure 2a). After exposing transgenic lines to different stress treatments, the GUS signal detected in the treatment group was significantly stronger than that detected in the control group (Figure 2c), further indicating that OsMLP423 may be involved in multiple stress responses. The GUS signal was particularly strongly expressed in roots (Figure 2d), which was consistent with the prediction results of the website, indicating that OsMLP423 may improve osmotic stress by regulating root water uptake.

### 2.4. OsMLP423 Is Highly Induced by Various Stresses

To further confirm the effects of phytohormones and abiotic stresses on *OsMLP423* expression, qRT-PCR was applied to examine relative expression levels of *OsMLP423*. The relative expression levels of *OsMLP423* in overexpression lines were verified by qRT-PCR (Figure 3d). Compared with the gene expression data on the RiceGE (gene expression map) website, we found that both drought stress and salt stress treatments could induce the expression of *OsMLP423*. The expression of *OsMLP423* under different stress treatments was further analyzed by qRT-PCR, and it was found that *OsMLP423* had a higher expression level in response to ABA, salt, and drought stresses (Figure 3a–c).

### 2.5. Enhanced Tolerance of OsMLP423 Overexpression Transgenic Plants to Drought Stress

Given that *Os**MLP423* expression was induced by drought stress, to test its function to drought stress tolerance, wild-type and transgenic lines were treated with a 20% *w*/*v* PEG solution at seedling stage, and a few overexpressed plants were randomly selected to measure the plant heights after one week of treatment. Plant heights of transgenic and wild-type plants were not different under control conditions. However, after 7 days of the 20% *w*/*v* PEG treatment, the heights of transgenic lines OE6-4, OE11-1, and OE12-3 were significantly higher than the wild type (Figure 4a,b). The above-mentioned young seedlings were transferred to sandy soil for 2 weeks, and then watering was stopped for 5 days (leaves started curling), and then they were returned to a normal water supply for 12 days. Compared with the wild type, overexpression lines showed less severe symptoms of drought stress, with delayed and less leaf curling. After the restoration of watering, the survival rate of wild-type plants was 32% and the overexpression lines had an average survival rate of 60% and also showed a lower water loss rate than the control (Figure 4c–f).

### 2.6. Enhanced Tolerance of OsMLP423 Overexpression Transgenic Plants to Salt Stress

Given that *OsMLP423* expression was induced by salt stress, to test its function regarding salt stress tolerance, the overexpression and wild-type lines were treated with a 150 mM sodium chloride solution, and the control group was treated with a standard solution without NaCl. After 7 days of treatment, we observed that the heights of the overexpression lines were similar to that of the wild-type line under normal conditions. However, after the 150 mM NaCl treatment, the plant heights of the overexpression lines OE6-4, OE11-1, and OE12-3 were significantly higher than that of the wild-type line (Figure 5a,b). After the NaCl treatment, plants were transferred to the standard solution without NaCl for 10 days, and the average survival rate (63%) of the overexpressing lines was significantly higher than that of the wild type (40%) (Figure 5c–e).

### 2.7. OsMLP423 Overexpression Lines Are Sensitive to ABA

To further confirm the role of *OsMLP423* in ABA-dependent stress responses, the sensitivity of overexpression lines to ABA was examined by analyzing seed germination and seedling growth. We statistically analyzed the germination rates of wild-type and overexpression lines after a 50 μM ABA treatment, and we carried out more than three biological repetitions. The germination rate of the wild type was 90%, while the germination of seeds of overexpression lines was delayed by 3 days and the germination rate was significantly lower (Figure 6a–c). Overexpression lines in the early stage (bud length about 2–3 mm) were treated with 50 μM ABA, and a standard nutrient solution without ABA was used in the control group. After 9 days, we randomly selected several transgenic seedlings to measure plant heights. Under normal conditions, the heights of overexpressing plants were similar to that of the wild type. However, after the 50 μM ABA treatment, the plant heights of transgenic plants were significantly lower than that of wild type (Figure 6d,e). These results indicated that overexpression of *OsMLP423* may have enhanced the sensitivity of rice to ABA. Exogenous ABA resulted in slower seed germination. The assumed increase in endogenous ABA concentration activated the expression of downstream stress-related genes, and plants enhanced their stress resistance by reducing plant height. To further explore the role of *OsMLP423* in the ABA signaling pathway and abiotic stress responses, we analyzed changes in the expression levels of the ABA biosynthesis and degradation genes under 50µM ABA using RT-PCR. The expression levels of the synthetic genes *OsNCED1* and *OsNCED3* in overexpression plants were significantly upregulated, whereas the expression levels of the ABA-degrading genes *OsABA8ox2* and *OsABA8ox3* were significantly downregulated, compared with the wild type (Figure 7a–d). At the same time, the expression levels of the ABA biosynthesis genes *OsNCED3* and *OsNCED4* and the ABA-inducible genes *OsRAB16* and *OsLEA3* in transgenic plants were significantly higher than those in wild-type plants under drought and salt treatment conditions (Figure 7e–h). Dehydration stress usually promotes ABA production and activates ABA signal transduction [24]. Based on the above results, we concluded that the overexpression of *OsMLP423* enhanced the adaptability of plants to drought and salt stress in an ABA-dependent way.

### 2.8. Overexpression of OsMLP423 Affects ROS Accumulation and Scavenging under Different Stresses

To characterize whether ROS accumulation was altered in the *OsMLP423* overexpression lines, we compared the accumulation of reactive oxygen species (ROS) between *OsMLP423* overexpression lines and the wild type under drought and salt treatments, and we assessed the accumulation of superoxide anion (O^2−^) and hydrogen peroxide (H_2_O_2_) using DAB and NBT staining, respectively. Under normal conditions, there was no obvious difference between wild-type and transgenic plants. After drought and salt treatments, DAB-stained transgenic leaves had lighter surface browning compared to the control (Figure 8a), and NBT-stained leaves had fewer surface spots than the wild type (Figure 8b). These further indicated that the transgenic plants produced less reactive oxygen species than wild-type plants under drought and salt stresses.

In addition, the activities of SOD, POD, and CAT under salt and drought stresses were detected. The results showed that there was no significant difference in the activities of superoxide dismutase, peroxidase, or catalase between the overexpression lines and the wild type under the control conditions, whereas under the salt and drought stress conditions, the overexpression lines exhibited significantly greater enzymatic activity than the wild type (Figure 9a–c). We also found that the content of MDA in the overexpression lines was significantly lower than that in the wild type, indicating that the *OsMLP423* overexpression lines had less cell membrane oxidative damage under stress conditions (Figure 9d).

## 3. Materials and Methods

### 3.1. Plant Materials and Bacterial Strains

The plant material used in this experiment was *Oryza sativa* L. subsp. *japonica* cv. Nipponbare. The strains used were *Escherichia coli* strain DH5a and *Agrobacterium* strain EHA105. The rice plants were grown in artificial climate incubator at 22 to 28 °C in a cycle of 16 h light/8 h dark [25]. After 2 weeks of cultivation, uniform seedlings were selected and subjected to the following abiotic stress treatments: salinity (150 mM), drought (20% *w*/*v* polyethylene glycol), and abscisic acid (50 µM).

### 3.2. RNA Extraction and cDNA Synthesis

After the above abiotic stress treatments of rice, fresh leaves were cut and quick-frozen in liquid nitrogen. The total RNA was extracted using Trizol reagent according to the manufacturer’s protocol [26]. The reverse transcription was conducted using a PrimeScript™ RT Reagent kit with a gDNA Eraser kit, and the cDNA was stored at −20 °C.

### 3.3. RT-PCR

Total RNAs of all the materials harvested were extracted using the Trizol reagent (Invitrogen) according to the manufacturer’s instructions. The extracted cDNA was used as a template to analyze the gene expression of *OsMLP423* under normal and abiotic stresses conditions. The primers for *OsMLP423* and the reference gene ubiquitin were designed using Primer Premier 5.0. The rice β-actin gene was used for internal control. The primers for *OsMLP423* gene are listed in Appendix A [27,28,29,30].

### 3.4. Analysis and Cloning of the OsMLP423 Gene

In order to construct an overexpression vector of *OsMLP423*, its open reading frame was cloned into the pCAMBIA1301S vector (from Huazhong Agricultural University), and gene expression was initiated under the control of CaMV 35S promoter. The connected clone vector was transferred to *E. coli*, screened, and cultured on solid LB plates containing kanamycin. The white colony was selected for colony PCR and was cut with Hind III and BamH I and sent for sequencing. The PCR primers are listed in Appendix A.

### 3.5. Analysis and Cloning of Promoter

A sequence of about 1400 bp before the translation initiation codon (ATG) of rice *OsMLP423* gene was obtained by comparison with the National Center for Biotechnology Information (NCBI) database, and the collected data were analyzed for potential regulatory elements. The polymerase chain reaction was used to amplify the promoter sequence of about 1400 bp upstream of *OsMLP423*, and the target fragment was fused to the GUS reporter gene in pCAMBIA1305. Then, the constructed vector was transferred into callus from wild-type plant via Agrobacterium-mediated transformation to obtain transgenic plants. All primers for the assays are listed in Appendix A.

### 3.6. Histochemical Detection of GUS Activity in Plants

The method described by Jefferson was used to detect GUS activity using histochemical staining. Different tissues of *OsMLP423pro:GUS* transgenic rice were placed in a buffer containing 50 mM NaPO_4_ (pH 7.2), 5 mM K_3_Fe (CN)_6_, 5 mM K_4_Fe(CN)_6_, 0.1% (*w*/*w*) Triton-100, and 1 mm X-Gluc, and they were incubated overnight at 37 °C [31]. The tissues were then soaked in 70% (*v*/*v*) ethanol for 5 min to stop the staining; then, 95% (*v*/*v*) ethanol was added and boiled until the chlorophyll was completely removed. Finally, photos were taken with a ZEISS stereo microscope.

### 3.7. Subcellular Localization of OsMLP423

The *OsMLP423* gene was linked to the pCAMBIA2300-eGFP vector (provided by the research group of Teacher Deng from the Rice Research Institute of Sichuan Agricultural University). The successfully ligated vector was transformed into Agrobacterium EHA105 and stored at −80 °C. The 35: GFP served as the control group. Forty-eight hours after tobacco transformation, observations were made using a laser confocal microscope [32]. All primers for the assays are listed in Appendix A.

### 3.8. Transgenic Plants Treated with Abiotic Stress

Wild-type and overexpression seeds were sterilized with 2% (*v*/*v*) NaClO solution for 30 min and were then incubated at 28 °C with a relative humidity of 70% and 14 h light/10 h dark cycle. The sterilized seeds were placed in 50 μM ABA solution for germination, and the germinated seeds were counted every day until the seventh day, and the control group was placed in the solution without ABA. The early seedlings (bud length of about 2 mm) were placed in 50 μM ABA solution for 9 days and 20% (*w*/*v*) PEG and 150 mM NaCl solution for 7 days, and the height of each plant was measured. Finally, the seedlings were cultured in sandy soil for 14 days; then, watering was stopped for 10 days to simulate field drought until the leaves curled, followed by recovery with normal watering for another 5 days to determine the survival rate.

The water loss rate measurement was conducted as described previously [33]. The leaves of 14-day-old rice seedlings were detached and exposed to air at room temperature and weighed at designated times. Water loss rate was calculated by reducing fresh weight with the initial time. Five leaves were collected from each line, and this was repeated three times.

### 3.9. Measurement of the Physiological Parameters

The two-week-old seedlings were used for two different treatments: drought treatment for 5 days in the soil and 150 mM sodium chloride treatment for 48 h. After completion of treatments, the physiological parameters of plants were determined. We measured the activity of antioxidant enzymes superoxide dismutase (SOD), peroxidase (POD), and catalase (CAT) [34]. The contents of malondialdehyde (MDA) were measured using a spectrophotometer as described elsewhere [35]. Leaves were placed in 1 mg/mL DAB and 6 mM NBT staining solution and incubated at 28 °C for 10 h in light. Anhydrous ethanol was used to remove chlorophyll. The accumulation of hydrogen peroxide and superoxide anion O^2−^ was observed under a stereo microscope.

### 3.10. Statistical Analysis

The data were analyzed by analysis of variance using SPSS statistics program. The statistical difference was clarified through analysis of variance by t-test, with *p* < 0.05 (*) and *p* < 0.01 (**) as significantly different.

## 4. Discussion

Rice is the main food source for more than half of the world’s population. Most rice varieties are severely damaged by abiotic stresses, which has severe social and economic impacts [36]. As a water-intensive crop, rice, unfortunately, makes its production challenging when encountering unexpected drought. As global temperature keeps rising, caused by climate change, it accelerates evaporation, which makes the soil suffer more droughts and become saltier [37]. This poses a challenge to researchers in improving water management systems and rice tolerance [38,39]. Therefore, to understand the response of rice to stress, it is particularly important to characterize potential candidate genes for the development of drought-resistant transgenic rice varieties for improved tolerant transgenic rice varieties.

The functions of the MLP protein family involved in abiotic stress responses and the regulation of plants have been reported in *cotton*, *tobacco*, *Arabidopsis*, and other organisms, but there is no research on its participation in rice stress resistance at present [40,41,42]. Bioinformatics and RT-qPCR analysis results showed that *OsMLP423* was strongly induced by drought and salt stresses. Therefore, we preliminarily determined that *OsMLP423* is a multistress response gene. To characterize the physiological function of *OsMLP423*, we constructed *OsMLP423* overexpression transgenic plants. The subcellular localization results showed that *OsMLP423* was located in the nucleus and cell membrane, similar to the localization results of its homologous genes. This shows that MLP proteins may play a similar role in different species. *OsMLP423* was mainly expressed in roots but also fully expressed in leaves and glumes in the later stage of development, which indicates that *OsMLP423* may be involved in the regulation of different stages of rice growth.

We found that *OsMLP423* overexpression lines had enhanced tolerance to drought and salt stresses compared with the wild type. When exposed to the osmotic adjustment substance PEG or a water deficit, overexpression lines showed greater tolerance to water stress and had significant plant heights, survival rates, and water retention statuses [43]. It was reported that overexpression in transgenic tobacco plants of the major latex-like protein gene *NtMLP423* resulted in greater sensitivity to abscisic acid (ABA)-mediated seed germination and ABA-induced stomatal closure, and it enhanced drought tolerance in tobacco by increasing endogenous ABA levels [44]. Ectopic overexpression of the cotton *GhMLP28* gene in *Arabidopsis* enhanced tolerance to salt stress [45]. All the aforementioned studies showed that MLP protein is involved in abiotic stress regulation, and our research shows similar results. The efficiency of water loss is related to drought stress tolerance, as more water loss rapidly increases sensitivity to drought. Our result showed that under drought stress, the *OsMLP423* overexpression lines showed less water loss compared with the wild type. Therefore, we speculated that under drought stress, *OsMLP423* induced stomatal closure of overexpressed plants through an ABA-dependent pathway, thus reducing water loss and enhancing drought resistance.

Since ABA mediates so many stress responses, the initial perception of dehydration and the subsequent changes in gene expression that lead to rapid ABA biosynthesis constitute the most important stress signal transduction pathway for plant responses to stresses [46]. Abscisic acid participates in plant development and regulates the adaptive response of plants to drought and high salinity [22,47]. Current models indicate that the abscisic acid-dependent pathway operates through ABRE, MYCRS, MYBRS, or NACRS cis-acting elements, while the abscisic acid-independent pathway operates using DRE/CRT or NACRS elements [48]. A promoter analysis showed that the promoter regions of all ABA-regulated genes contain the ABREcis motif [47]. Our analysis found that ABA-inducing elements also exist in the upstream promoter region of the *OsMLP423* gene. In addition, it also contains MBS (MYB binding site involved in drought response induction). Our quantitative analysis showed that *OsMLP423* was strongly induced by 50 μM ABA, and the expression level of *O**sMLP423* increased by more than 4 times after 16 h compared with the control. The root GUS activity test also showed that the root signal after ABA treatment was significantly stronger than that of the control. Both *OsMLP423* and ABA receptors PYL/PYR belong to the Bet v1 family and share a similar protein structure [42], so we speculated that *OsMLP423* might be involved in the ABA signaling pathway. In subsequent experiments, we found that the germination and seedling growth of *OsMLP423* overexpression lines were severely inhibited in the presence of ABA. We speculated that *OsMLP423* may be involved in the ABA signal transduction pathway to affect rice germination and early growth. To further elucidate this point, comparisons of the expression levels of ABA-responsive genes, *OsNCEDs* and *OsABA8oxs*, were performed between the normal and ABA treatment conditions. They showed that a higher expression level of ABA synthetic genes, *OsNCED1* and *OsNCED3*, and a lower expression level of ABA-degrading genes, *OsABA8ox2* and *ABA8ox3*, were detected in overexpression lines under ABA treatment compared with the wild type. These results showed that *O**sMLP423* may regulate ABA signaling and ABA accumulation by regulating ABA biosynthesis. Meanwhile, we analyzed the expression of ABA synthesis genes and stress-induced marker genes that play a role in the ABA-dependent pathway under drought and salt conditions. The expressions of the ABA biosynthesis genes *OsNCED3* and *OsNCED4* and the ABA-inducible genes *OsRAB16* and *OsLEA3* were higher in overexpression lines compared with the wild type. Under drought stress, cells can quickly induce ABA synthesis, thus slowing down the transpiration rate of plants [49,50,51]. Therefore, we conclude that *OsMLP423* regulates the expression of downstream stress-related genes by increasing endogenous ABA content to further enhance drought and salt tolerance in transgenic plants.

Abiotic stresses are usually accompanied by reactive oxygen species (ROS) accumulation, which subsequently activates antioxidant defense systems and then confers stress resistance [52]. After the drought and salt stress treatments, DAB and NBT staining results also showed that overexpressed lines accumulated less H_2_O_2_ and O^2-^. These results indicated that overexpressed *OSMLP423* rice plants enhanced their tolerance to oxidative stress.

Accumulating evidence indicates that ABA-enhanced water stress tolerance is associated with the induction of antioxidant defense systems, including ROS-scavenging enzymes such as SOD, CAT, and POD [53]. Under stress conditions, the SOD, POD, and CAT activities of transgenic plants can reflect the impact of stress on plants to a certain extent [54,55]. In the present study, there were no significant differences in physiological indicators between transgenic and wild-type seedlings under normal growth conditions. However, under drought and salt stress, the activities of SOD, POD, and CAT in *OsMLP423*-overexpressing plants were significantly increased. This suggests that *OsMLP43* may play a role in scavenging the excessive ROS production under dehydration to enhance drought tolerance. Taken together, these findings strongly suggest that overexpression of *OsMLP423* could increase the activity of ROS-scavenging enzymes and reduce cell damage under different stresses. MDA was measured as an index of oxidative damage in cells. Under salt and drought stress, the MDA content of *OsMLP423* overexpression lines was significantly lower than that of the WT, which indicated that the oxidative damage in the overexpression cells was also lower.

## 5. Conclusions

In summary, we screened a stress-related gene, *OsMLP423*, induced by multiple stresses from the transcriptome library of rice drought treatment, and the *O**sMLP423* was located in both the nucleus and membrane system. Our experimental results show that *OsMLP423*-overexpressing plants are more tolerant to drought and salt but very sensitive to ABA. These results provide basic research on the *OsMLP423* gene in plant responses to abiotic stress and provide materials for developing potential candidate genes of drought-tolerant transgenic rice varieties.

## Figures and Tables

**Figure 1 plants-11-01653-f001:**
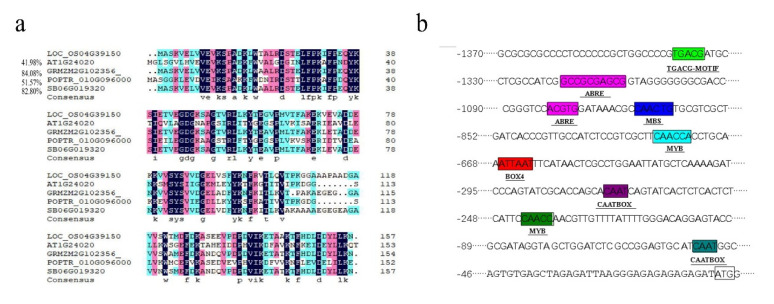
Bioinformatics analysis of *OsMLP423*. (**a**) Analysis of homologous amino acid sequence of *OsMLP423*. (**b**) Analysis of the relevant elements of the promoter region.

**Figure 2 plants-11-01653-f002:**
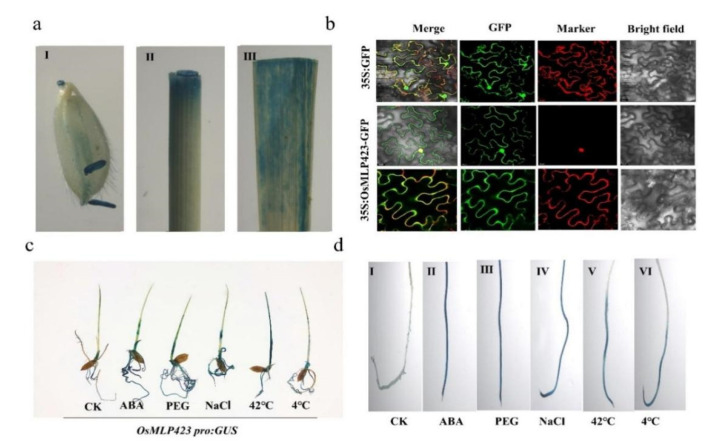
Expression pattern analysis of *OsMLP423*. (**a**) *OsMLP423pro:*GUS transgenic plants showed staining of different tissues. (I: glumes; II: stems; III: mature leaf.) (**b**) Subcellular localization of OsMLP423 in tobacco. Scale bars are 20 mm. (**c**) GUS staining of *OsMLP423pro*:GUS transgenic plants under different stress treatments. (**d**) Detection of GUS activity in roots of OsMLP423 promoter under abiotic stress. (Ⅰ: CK; Ⅱ: ABA; Ⅲ: PEG-6000; Ⅳ: NaCl; Ⅴ: 42 °C; Ⅵ: 4 °C).

**Figure 3 plants-11-01653-f003:**
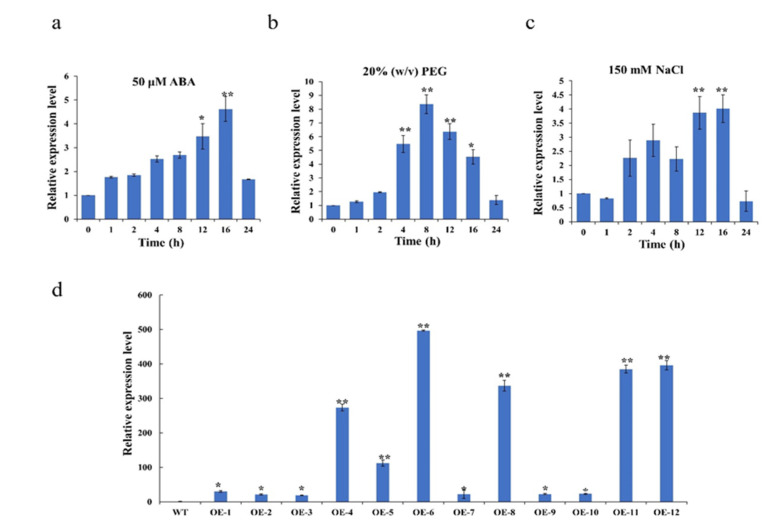
Expression analysis of *OsMLP423* under different stresses and strains selection. (**a**–**c**) RT-PCR analysis of *OsMLP423*: (**a**) 50 μM ABA; (**b**) 20% *w*/*v* PEG; (**c**) 150 mM NaCl. (**d**) Expression levels of *OsMLP423* transgenic lines. Values are mean ± SE (*n* = 3). Asterisks indicate significant differences between transgenic lines and WT (* *p* < 0.05, ** *p* < 0.01). *n* = 20 plants per treatment.

**Figure 4 plants-11-01653-f004:**
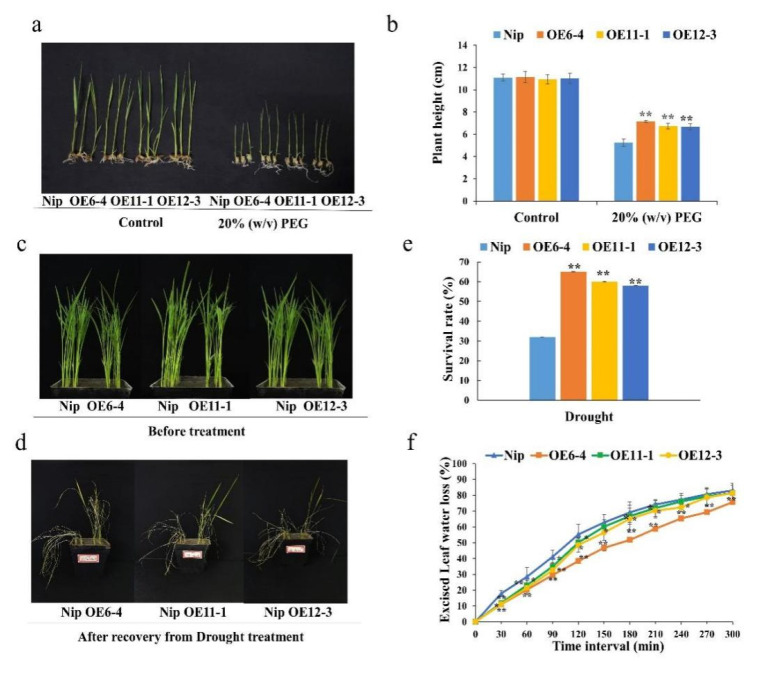
Drought stress treatment of *OsMLP423* transgenic lines. (**a**) *OsMLP423* transgenic plants grown at 20% *w*/*v* PEG for 7 days. (**b**) Relative plant heights. (**c**,**d**) Phenotypic differences between *OsMLP423* overexpressing and wild-type plants during recovery after drought treatment. (**e**) Survival rate. (**f**) Water loss rates of detached leaves from 14-day-old plants. Values are mean ± SE (*n* = 3). Asterisks indicate significant differences between transgenic lines and WT (* *p* < 0.05, ** *p* < 0.01). *n* = 20 plants per treatment.

**Figure 5 plants-11-01653-f005:**
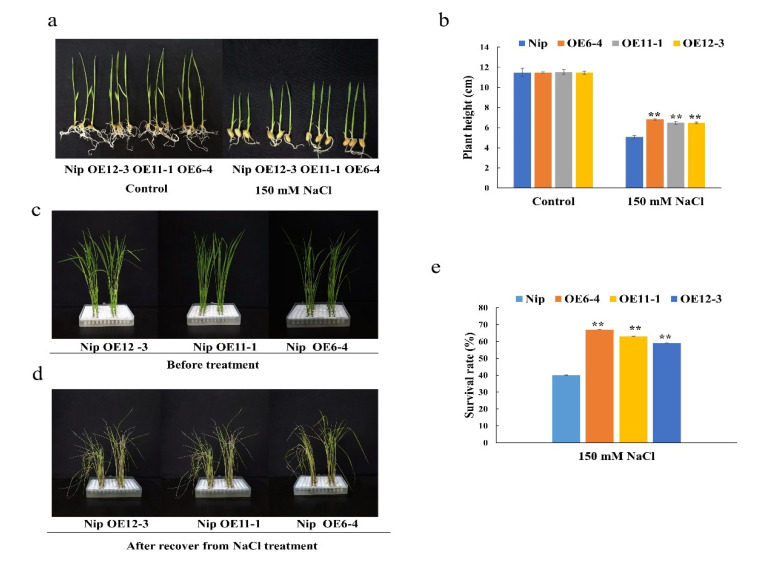
Salt stress treatment of *OsMLP423* transgenic lines. (**a**) *OsMLP423* transgenic plants after 150 mM NaCl for 7 days. (**b**) Relative plant heights. (**c**,**d**) Phenotypic differences in *OsMLP423* overexpression and wild-type recovery after 150 mM NaCl treatment. (**e**) Survival rate. Values are mean ±SE (*n* = 3). Asterisks indicate significant differences between transgenic lines and WT (** *p* < 0.01). *n* = 20 plants per treatment.

**Figure 6 plants-11-01653-f006:**
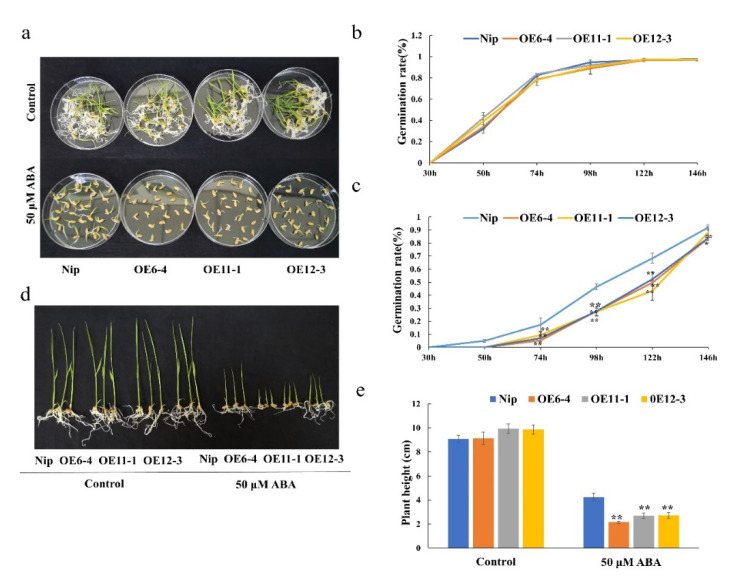
ABA treatment of *OsMLP423* transgenic lines. (**a**) Germination of *OsMLP423* overexpression lines treated with 50 μM ABA for 7 days. (**b**) Line graph of germination rate of *OsMLP423* overexpression plants in normal water culture for 7 days. (**c**) Line graph of germination rate of *OsMLP423* overexpressing plants treated with 50 μM ABA for 7 days. (**d**) *OsMLP423* transgenic plants treated with 50 μM ABA for 9 days. (**e**) Relative plant heights. Values are mean ± SE (*n* = 3). Asterisks indicate significant differences between transgenic lines and WT (* *p* < 0.05, ** *p* < 0.01). *n* = 20 plants per treatment.

**Figure 7 plants-11-01653-f007:**
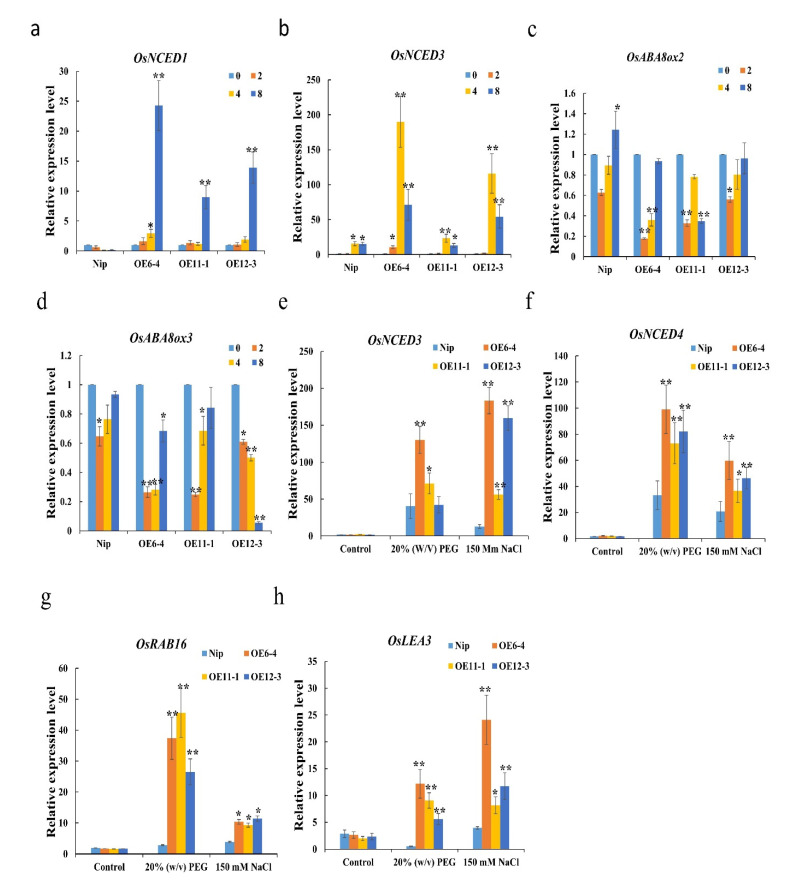
Expression analysis of ABA-related genes under different stress conditions. (**a**–**d**) The analysis of the expression levels of *OsNCED1, OsNCED3*, *OsABA80x2,* and *OsABA80X3,* respectively, under 50 μM ABA treatment. (**e**–**h**). The analysis of the expression levels of *OsNCED3*, *OsNCED4*, *OsRAB16,* and *OsLEA3*, respectively, under 20% *w*/*v* PEG and 150 mM NaCl stress. Leaf samples were collected during 24h treatment. Values are mean ± SE (*n* = 3). Asterisks indicate significant differences between transgenic lines and WT (* *p* < 0.05, ** *p* < 0.01). *n* = 20 plants per treatment.

**Figure 8 plants-11-01653-f008:**
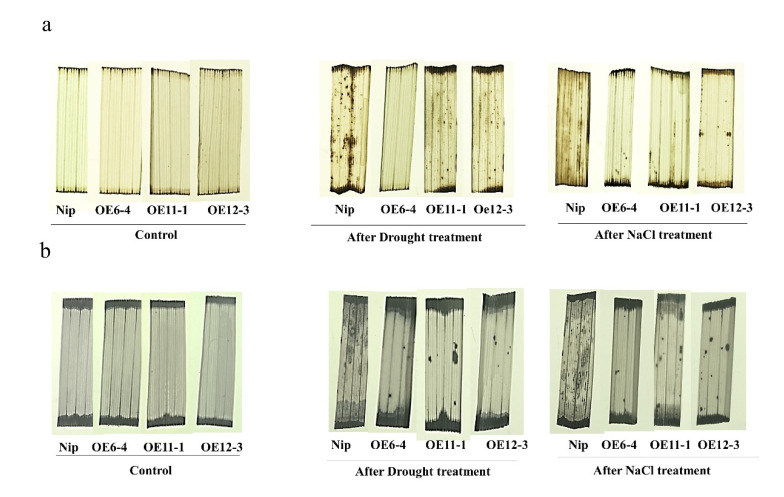
Accumulation of reactive oxygen species in *OsMLP423* transgenic plants. (**a**) DAB staining was used to detect superoxide anion (O^2−^) in the plants overexpressing *OsMLP423* before and after drought and salt treatments. (**b**) Hydrogen peroxide accumulation in plants overexpressing *OsMLP423* before and after drought and salt treatments was detected using NBT staining.

**Figure 9 plants-11-01653-f009:**
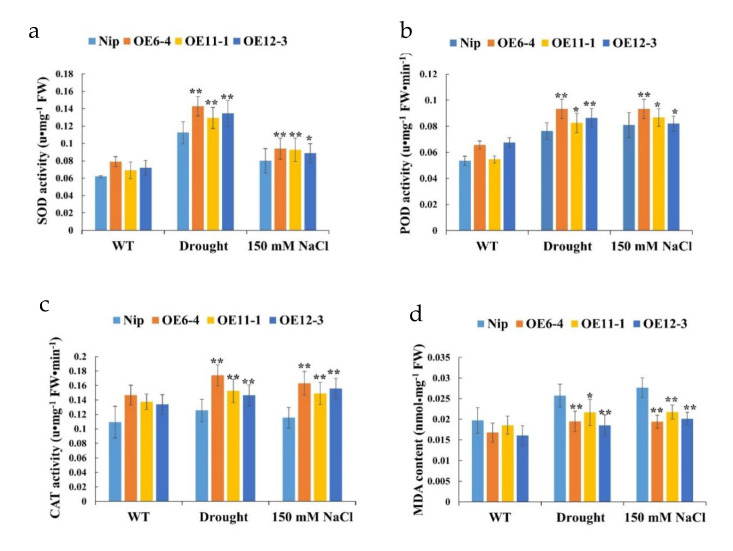
Physiological activity of WT and transgenic plants in response to drought and salt stresses. (**a**) SOD activity. (**b**) POD activity. (**c**) CAT activity. (**d**) MDA content. WT: wild-type rice; OE6-4, OE11-1, and OE12-3 are three independent transgenic lines. Values are mean ± SE (*n* = 3). Asterisks indicate significant differences between transgenic lines and WT (* *p* < 0.05, ** *p* < 0.01). *n* = 20 plants per treatment.

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
