# Peer review of "OsMLP423 Is a Positive Regulator of Tolerance to Drought and Salt Stresses in Rice"

_plants, 2022, doi:10.3390/plants11131653_

Round 1

Reviewer 1 Report

The manuscript entitled OsMLP423 is a positive regulator of tolerance to drought and salt stress in rice" is based on original research experiment and the presented results therein broaden the knowledge of plant sciences. Authors screened a drought-inducing gene containing about 158 amino acid residues from the transcriptome library of rice exposed to drought treatment, and found ABA-related cis-acting elements and multiple drought-stress-related cis-acting elements in its promoter sequence. There is no doubt that this work is in the scope of Plants journal. The publication presents some interesting studies. The work delivers some interesting results and can be important source of valuable information.

The introduction is properly composed. The materials and methods section contains the basic requested elements and provide information about the experimental preparations, analyses. However, details about growth conditions were omitted. The data analysis is generally properly provided. The results show valuable information. The obtained data are discussed sufficiently.

However, the authors made many shortcomings that must be corrected before the publication of the work:

1) Abstract: There is no aim of the work. There is also no conclusions of results. the reader will not find out what mechanism the authors described in the work.

2) Key word: should not be repeated with the words used in the title.

3) Authors did not make a research hypothesis. In fact, it is not known what mechanism of the factors the authors want to explain.

4) All units should be in SI format. So, no mg/100g but mg 100 g-1 for example.

5) Under what conditions was the experiment conducted: was it a growth chamber or a phytotron?

6) Looking at the germination results, I am sure that statistics should not be used here because there are no repetitions. The authors assumed that each germinated seed is a repetition, but this is not true. This is the population. The dishes are a repetition in such experiments. And here is one pan per variant. Not only that - 20 seeds is a very small population. Typically, a population of 50 or 100 seeds is made.

7) the layout of the article is incorrect (please see the journal's website). The MM section should be discussed. There are no conclusions at all (which is a part of the work).

Author Response

Dear reviewer:

We have responded to your comments one by one, please check the attachment.

Reviewer 2 Report

i suggest major revision

Author Response

(The authors gave the same response as above.)
